# Novel Therapeutic Approaches for Mitigating Complications in Short Bowel Syndrome

**DOI:** 10.3390/nu14214660

**Published:** 2022-11-04

**Authors:** Jeffery Bettag, Loren Po, Cassius Cunningham, Rahul Tallam, Kento Kurashima, Aakash Nagarapu, Chelsea Hutchinson, Sylvia Morfin, Mustafa Nazzal, Chien-Jung Lin, Amit Mathur, Rajeev Aurora, Ajay K. Jain

**Affiliations:** 1Department of Pediatrics, Saint Louis University School of Medicine, Saint Louis, MO 63103, USA; 2Department of Surgery, Saint Louis University School of Medicine, Saint Louis, MO 63103, USA

**Keywords:** short bowel syndrome, parenteral nutrition, gut-liver axis, gut brain axis, FXR, FGF19, GLP, Omega 3, lipids

## Abstract

Short bowel syndrome (SBS) is a particularly serious condition in which the small intestine does not absorb sufficient nutrients for biological needs, resulting in severe illness and potentially death if not treated. Given the important role of the gut in many signaling cascades throughout the body, SBS results in disruption of many pathways and imbalances in various hormones. Due to the inability to meet sufficient nutritional needs, an intravenous form of nutrition, total parental nutrition (TPN), is administered. However, TPN presents difficulties such as severe liver injury and altered signaling secondary to the continued lack of luminal contents. This manuscript aims to summarize relevant studies into the systemic effects of TPN on systems such as the gut–brain, gut-lung, and gut-liver axis, as well as present novel therapeutics currently under use or investigation as mitigation strategies for TPN induced injury.

## 1. Introduction

Short bowel syndrome (SBS) is a devastating condition characterized by an inability of the small intestine to effectively absorb adequate nutrients necessary for survival. SBS generally occurs secondary to either missing or damaged small intestine due to congenital abnormalities or surgical removal of a section of the small intestine. Surgically, SBS occurs in 15% of adult patients undergoing intestinal resection and congenitally or neonatally in 2.2% of newborns [1,2]. According to a retrospective review of 210 charts done in 2005 by Thompson, reoperation, cancer and irradiation, and Crohn’s disease were the most common conditions leading to SBS in adults [3]. The major symptoms of SBS commonly manifest due to a loss of absorptive area, but also may be caused by damage to nutrient transporters, endocrine cells and hormone signaling [1]. Symptomatology is also characterized by which area of the intestine is resected or missing due to the site specific nature of nutrient absorption [4,5,6]. Given that SBS is most common due to resection of the distal ileum and proximal colon, hormones produced in this area and their respective signaling pathways may be disrupted. Therefore, production and regulation of glucagon like peptides 1 and 2 (GLP1 and GLP2), neurotensin, and peptide YY are often deficient while hormones produced in the proximal intestine, such as gastrin, secretin, and motilin are typically intact with such etiology [5,6].

In the absence of sufficient ability to maintain nutritional needs during enteral feeding, SBS patients require an intravenous form of nutrition known as TPN for survival [1]. Clinically, SBS is associated with the need for TPN dependance for greater than 60 days after intestinal resection or a bowel length of less than 25% of expected [7]. SBS may become life threatening and progress towards disease complications such as chronic intestinal failure (CIF) and intestinal failure associated liver disease (IFALD). This is typically seen in conditions such as necrotizing enterocolitis, volvulus, and intestinal atresia or strictures [8]. However, most relevant to this review is the generalized gut atrophy secondary to the state induced by TPN, resulting from a lack of luminal contents [1,9,10]. Although the benefits of TPN are widely known and TPN is used as a life-saving treatment worldwide, longer durations of TPN can cause severe liver injury, and lack of luminal contents can induce serious and deadly complications [10,11]. Therefore, therapeutic strategies tend to focus on initiating enteral nutrition (EN) and enhancing intestinal adaptation [11,12]. Additionally, while working to restore enteral nutrition, common symptoms such as diarrhea, dehydration, nutrient imbalances, and weight loss may need to be managed through classical treatments such as antimotility agents, IV fluid replacement, serum testing and TPN adjustments, or hormone therapy [13]. Prognosis of treatment is heavily correlated with presence or absence of the ileocecal valve (ICV), needed for preventing bacterial overgrowth and regulating intestinal transit time. The status of the ICV is a strong predictor of the ability to wean off from TPN and achieve enteral autonomy, and its presence or absence is considered in prognosticating and treatment planning [14]. Overall, from promoting intestinal adaptation, testing serum nutrients and adjusting TPN, to surgical treatment and in rare cases intestinal transplantation, SBS is a complicated medical and surgical condition [14,15]. Although the etiology of TPN associated injury is not well known, several recent advancements have been made in understanding the mechanistic pathways. Ultimately, this literature review aims to understand and summarize what is known about the mechanisms of SBS and TPN that are currently utilized in diagnostic and therapeutic strategies to treat and mitigate associated complications and disease.

## 2. Alterations of Mechanical Signaling and Effects of TPN on the Body

### 2.1. Gut-Liver Axis

TPN is noted to have numerous effects on physiology across many organ systems in the body. Among the many systems, particularly important is the alteration in signaling within what is known as the gut-liver axis. It is now widely known that a lack of intestinal contents alter luminal signaling (Figure 1).

#### 2.1.1. Role of FGF19 on Bile Acid Synthesis

In healthy individuals, enterohepatic circulation includes activation of the nuclear receptor, Farnesoid X Receptor (FXR), in the gut by luminal bile acids resulting in the release of Fibroblast Growth Factor 19 (FGF19). FGF19 is known to regulate bile acid synthesis. However, this process is inhibited during TPN due to a lack of FGF19 that drives suppression of intrahepatic cholesterol 7 alpha-hydroxylase (CYP7A1), which is a rate limiting step in bile acid synthesis. This pathway accounts for the majority of bile acid synthesis [16]. FGF19 is expressed in the distal small intestine and its synthesis depends on the activation of FXR which occurs normally in the presence of intestinal food and bile acids. Therefore, FGF19 levels increase following an enteral feed which would inhibit CYP7A1, however during TPN FGF19 is decreased due to a lack of luminal contents [17]. Using a piglet model, results indicate that enteral administration of a strong FXR agonist, chenodeoxycholic acid (CDCA), prevents cholestasis in the context of parenteral associated liver disease (PNALD). Thus, it is predicted that TPN induced pathologies such as PNALD are caused by the disruption of enterohepatic circulation and the FXR-FGF19 signaling axis [18].

#### 2.1.2. TGR5-GLP Pathway

TGR5 is a G-protein coupled receptor for bile acids that regulates glucose homeostasis, lipid, and energy metabolism [19]. TGR5 is activated by several bile acids, with lithocholic acid (LCA) as the highest naturally potent agonist. TGR5 is found in enteroendocrine L cells in the lumen of the small intestine, and its activation leads to systemic release of several hormones, notably glucagon-like peptide (GLP) 1 and GLP2. The state of TPN drives the impaired gut-liver signaling across the TGR5/GLP1/2 system as with the FXR/FGF19/FGR4 axis, which leads to liver injury including hepatic steatosis and liver fibrosis. TGR5 also reduces the inflammatory response via the TGR5-cAMP-dependent pathway, as well as drives a reduction in the NF-κB-dependent inflammatory responses [20].

#### 2.1.3. Gut Mucosal Atrophy

Recent research hypothesized that TPN induced gastrointestinal mucosal atrophy may be caused by the absence of intraluminal contents which can alter associated regulatory signals [21]. TPN induced atrophy of the gut mucosa can occur quickly and is characterized by several morphological alterations that include decreased villous height, surface area, crypt depth, and epithelial cell count [22]. A high metabolic rate and rapid turnover of the intestinal epithelial cells without proper replacement leads to a reduced cell count [23]. Using a rat model, Sugita’s study explored the potential benefits of hepatocyte growth factor (HGF) on the complications presented from mucosal atrophy induced by TPN. Although the mechanism is unknown, It’s hypothesized that the preservation of the jejunal villus height can be attributed to the increased crypt cell proliferation rate in the jejunum. This is thought to be the mechanism because HGF could have induced elongation of the ileal mucosa leading to the higher crypt cell proliferation rates. The design of Sugita’s study consisted of four groups; mice fed orally, TPN only, TPN with a low dose of 0.3 mg/kg/day of HGF, and TPN with a high dose of 1.0 mg/kg/day of HGF. The results of this study suggested HGF significantly prevented the mucosal atrophy as seen through increased jejunal villus height in the HGF treated groups compared to that on TPN, as well as an increase in the crypt cell proliferation rates in the jejunum. However, there was no significance in the absorptive mucosal area between the HGF treated mice compared to the TPN group. This could potentially stem from insufficient dose of HGF, which would require further research to investigate the optimal dosage [24]. Due to the attenuation of a few complications characterized by mucosal atrophy we can hypothesize that HGF may play a key role in mitigating gastrointestinal mucosal atrophy caused by TPN.

#### 2.1.4. Gut Microbiota

Due to a lack of luminal nutrition, data suggests that there are significant alterations in gut microbiota as well as gut derived signaling [18,25,26,27]. It has been documented that there is dominance of the Firmicutes phylum in normal individuals. In SBS there is a proliferation of the Bacteroidetes phylum [25,26]. Using multilevel logistic regression, significant sub-phylum changes in microbial community composition have been noted with use of TPN [28,29]. Pertinently, within the Bacteroidetes phylum there is proliferation of organisms belonging to the Porphyromonadaceae family (class Bacteroidetes, order Bacteroidales). Gram-negative bacterial families such as Bacteroidaceae, Fusobacteriaceae, Campylobacteraceae and Burkholderiaceae are associated with increased LPS and inflammatory cytokines, during TPN.

Data also suggest that such microbial shifts promote intestinal inflammation and increase intestinal permeability [30,31,32,33].

Indeed, it has been hypothesized that gut microbial alterations enable bacterial flux across the mucosa which results in cytokine mediated hepatocellular injury [34,35,36]. Upon initiation of enterally administered antibiotics, a significant reduction in liver injury has been reported [37,38,39]. Additionally, significantly elevated Interleukin-6 (IL-6) and tumor necrosis factor-α (TNF-α) have been noted in SBS [40,41,42].

### 2.2. Gut-Brain Axis

A lack of luminal content, with nutrition replaced by IV fluids as in TPN, changes the physiology of glucose metabolism in the body (Figure 2). Notably, glucose serves as the brain’s primary source of nutrition, and glucose, along with many other nutrients, serves important neuroprotective roles. Thus, an alteration in glucose signaling could affect cognition in adults, and more devastatingly neurodevelopment in infants.

In a study comparing rats on TPN vs. enterally fed rats, both given access to regular chow, at the end of a two-week experiment, brain glycogen and triglyceride levels were recorded. The authors noted that TPN rats had an 85% decrease in food intake, and a 23% increase in brain glycogen, indicating a connection between the amount of glycogen in the brain and drive for seeking enteral nutrition [43]. This was also reflected by an increase in glycogen phosphorylase and a concurrent decrease in glycogen synthase. In a separate study on TPN fed rabbits by Lopez and Rassin, amino acids, particularly tyrosine, were found to be decreased in the brain at the end of a 7-day period [44]. In the same study, serum values of large amino acids were correspondingly higher due to the selectivity of the blood–brain barrier. Radmacher independently proposed a solution to this finding by noting increases in tyrosine levels by a method of altering the tyrosine in TPN formulation to glutamyl-tyrosine [45]. Brain fatty acid composition has previously not been shown to change as a result of TPN [46], however recent studies have found some differences in specific lipids [11]. Lee et al. further investigated the effect of TPN on feeding behavior. Neuropeptide Y and cholecystokinin are peptides known to be involved in the regulation of satiety in the brain through a gastrohepatic caloric sensory feedback loop [47]. Lee found that neuropeptide Y was significantly elevated in TPN control rats after 72 h when compared to enterally fed rats [48]. A recent systematic review of 12 randomized controlled trials and 23 cohort trials on the effects of TPN on neurodevelopment done in 2022 by De Nardo et al. revealed that there was no effect on cerebral growth via MRI and that no trials made any significant conclusion about variations in early energy intake on neurodevelopment [49]. There is ongoing work addressing the impact of mental illnesses with SBS as well as the use of TPN, which remains an important ressearch focus [50,51]. Overall, there appears to be a relative lack of studies into the effect of TPN, especially in the context of SBS, on both localized and whole brain function.

### 2.3. Gut-Lung Axis

Much like the brain, TPN associated lung injury is frequently reported but not well mechanistically understood. Long term TPN use is known to be linked to an increased incidence of bronchopulmonary dysplasia and pneumonia in newborns due to suppression of autophagy and lung barrier impairment (Figure 2) [52]. TPN can induce apoptosis of cells in the lamina propria of the intestines, reduce secretions of IgA, and impair microbiota and barrier functions. Because the lung is also a luminal organ and shares a common embryological origin, similar effects have been investigated as what is known as “gut-lung crosstalk” [53]. Lung inflammation associated with inflammatory bowel diseases has been proposed to accompany the allergic reaction of the bowel in what is commonly referred to as the “social property of cells” [54]. For instance, Zhang et al. found that ovalbumin sensitization of lung cells drives lung cells to produce TSLP4 (thymic stromal lymphoprotein 4), resulting in the development and maintenance of asthma [55]. Donovan et al. also found a similar relationship showing that a well-known gut inflammasome, nucleotide-binding oligomerization domain (NOD)-like receptor (NLR) family, pyrin domain-containing protein 3 (NLRP3) inflammasome, causes inflammatory responses in the lungs when activated by damage associated molecular patterns or in responses to microbiota changes, changes which also occur with TPN [56]. Additionally, TPN resulted in weakening of tight junctions in pulmonary alveolar epithelium, resulting in inflammation and apoptosis [57]. TPN also depletes pulmonary lymphocyte populations owing to a reduction in autophagy [58]. As recently as April 2022, while investigating the effect of bronchopulmonary dysplasia on gut microbiota, Fan et al. showed that the gut lung axis is also bidirectional. Lung tissue showed change prior to changes in the microbiota of mice in hyperoxic states by generation of TNF-α, IL-6, IL-8 and MCP-1 [59]. The close interaction between respiratory and gastrointestinal systems during disease states and dysbiosis is well documented, but the understanding of mechanical pathways and treatment options lack thorough investigation.

## 3. Novel Therapeutics (Table 1)

### 3.1. GLP-1 and GLP-2

Long term TPN has several side effects, one of them being atrophy of mucosal villi [60]. As longer villi contribute to a larger absorptive surface their atrophy would decrease the intestine’s ability to absorb nutrients and increase the reliance on TPN [60]. Glucagon-like peptide-2 (GLP-2) is an adenylate cyclase-activating peptide secreted by ileal and colonic enteroendocrine L cells in response to both proximal enteric neuronal signaling and the presence of luminal nutrients [61]. Infusions of GLP-2 have been demonstrated in rat models to increase total small bowel weight, villus height, crypt depth, and total mucosal surface area. In states of increased nutrient delivery to the hindgut, from either increased nutrient availability or proximal intestinal resection or dysfunction, GLP-2 induces intestinal adaptation [62]. It has been used to improve both cholestasis and liver injury in piglet models [63]. Administration of low-dose GLP-2 (1 µg/kg/h) to parenteral fed rats has been used to improve hepatic steatosis in intestinal failure associated liver disease [64]. With these studies in mind, it is possible GLP-2 may have a similar effect on humans. GLP-2 analogs have also been used in humans with SBS. In a 24 week study such was successful at reducing the volume of parenteral support needed by the experimental group by 4.4 ± 3.8 L/wk as compared to the placebo groups reduction of 2.3 ± 2.7 L/wk. The study concluded that the GLP-2 analog was effective at reducing the volume and numbers of days of parenteral support needed for patients with SBS with intestinal failure [52]. It was found that the GLP-2 analog was most effective on patients who have had jejunostomies or ileostomies as this area of the bowl is where the L cells that secrete GLP-2 reside [65]. This treatment was found to be effective and safe in both pediatric and adult patients [66]. The long-term effects of a GLP-2 analog were investigated in a 30-month trial and found to be effective at continuing to reduce the need for parenteral nutrition support [67]. A long-acting GLP-2 analog was designed for once-weekly dosing and was tested in a placebo-controlled trial. Eight patients were given once-weekly 5-mg doses followed by a placebo for 4 weeks. This was then followed by a once-weekly 10-mg dose for 4 weeks. Urine volume and urinary sodium excretion were used as markers of functional intestinal rehabilitation. Treatment at both doses significantly increased urine output to a mean of 714 mL/day (95% CI, 490–939; *p* < 0.05) and 795 mL/day (95% CI, 195–1394; *p* < 0.05) respectively. Some side effects noticed were mild to moderate and included polyuria, decreased stoma output, stoma complications, decreased thirst, and edema. This trial suggested that such long acting GLP analogs are a possible treatment for SBS [68] (Table 1).

**Table 1 nutrients-14-04660-t001:** Therapeutics for SBS ^1^.

Therapeutic	Mechanism of Action	Overall Effect	Adverse Effects
GLP2	Adenylate cyclase activating peptide	↑ Total small bowel weight ↑ Villus height ↑ Crypt depth ↑ Total mucosal surface area	Polyuria Decreased stoma output Stoma complications Decreased thirst Edema
GLP1	Adenylate cyclase activating peptide Vagus nerve inhibition Decrease in gastric emptying, gastric acid secretion	↓ Fecal wet weight loss ↓ Energy loss ↓ Nitrogen loss ↓ Sodium loss ↓ Potassium loss	Not yet explored
Peptide YY	Secreted by ileal L cells	Not yet explored	Not yet explored
IGF-I	Increases protein synthesis in jejunal muscularis layer Increases mucosal DNA and protein	↑ Crypt depth ↑ Body weight ↑ Intestinal mucosal cellularity	Not yet explored
Ghrelin	Stimulates release of GH and IGF-1 Promotes intestinal epithelial cell proliferation through PI3K/Akt pathway EGFR trans-activation Ultimately converges to ERK ½ phosphorylation	↑ Villus height ↑ Crypt depth ↑ Crypt cell proliferation rate	Not yet explored
Thyroid Hormone	Trophic hormone Regulation of gene transcription Induction of intestinal alkaline phosphatase (IAP) mRNA	↑ Jejunum weight ↑ Ileum weight ↑ Jejunal villus height ↑ Crypt depth ↑ Enterocyte proliferation	Not yet explored
Cholecystokinin	Peptide hormone secreted by endocrine cells Stimulates gallbladder contraction Promotes small intestine motility	↓ Incidence of gallbladder stasis ↓ Gallbladder: hepatic bile 3H-cholic acid specific activity ratio ↓ Incidence of detrimental changes in bile composition and hepatic bile secretion ↓ IL-1 level ↓ TNF-alpha levels ↓ Bilirubin level ↑ Bile flow	Not yet explored
Omega 3	Polyunsaturated fat	↑ Bowel weight ↑ Mucosal weight ↑ Mucosal cDNA ↑ Protein expression ↑ Villous height ↑ Crypt depth ↑ Cell proliferation ↑ Apoptosis ↓ Bilirubin level	Not yet explored
TGR5 Agonist	Increases secretion of GLP	No significant improvement in growth or proliferation of cells	Not yet explored
Rapamycin	Immunosuppressive drug mTOR kinase inhibitor Prevents progression of cell growth	↓ Autophagy ↓ Reactive oxygen species ↓ Binding immunoglobulin protein (BIP) ↓ Spliced X-box-binding protein-1 (sXBP1) ↓ Steatosis↑ Hepatic Function	Not yet explored
Chenodeoxycholic Acid (CDCA)	Dual agonist of FXR and TGR5	↓ Direct bilirubin level ↓ Serum Triglyceride ↓ Bile acid ↑ Weight gain ↓ Incidence of pigmented gallstones	Not yet explored
Carbamazepine	Anti-epileptic Inhibition of voltage-gated sodium and calcium channels	↑ Clearance of misfolded proteins ↓ Hepatic Load ↓ Fibrosis ↓ Hepatic cholestasis deposits ↓ GGT in TPN	Elevated GGT Elevated ALT Elevated AST
Surgical Management	Liver, small intestine, or multivisceral transplantation Longitudinal intestinal lengthening and tailoring (LILT) Serial transverse enteroplasty (STEP)	Surgical Cure	Variable survival rate

1. ↑ indicates that the therapeutic causes increase in the subsequent factor and ↓ indicates the therapeutic causes a decrease in the following factor.

Like GLP-2, glucagon-like peptide-1 (GLP-1) is an adenylate cyclase-activating peptide. Along with other actions on pancreatic B-cells, GLP-1 acts on the vagus nerve to decrease gastric emptying and gastric acid secretion [69,70]. It has been theorized that part of the reason patients with SBS have malabsorption is the loss of GLP-1 controlled braking system in addition to the loss of absorptive surface area [20]. A GLP-1 analog was trialed in five patients with SBS. The results were promising as all patients had improvements in motility and three of the patients were able to be weaned off TPN [71]. A second study of nine patients with SBS reinforced this result. In this study nine patients with SBS were given infusions of GLP-1, a combination of GLP-2 and GLP-1+2, and a placebo at 1 pmol/kg/min. Results were gathered by measuring the difference between the weight of the oral intake and fecal output, measuring fasting body state and bone composition, and by analyzing blood samples that were taken at intervals after feeding for blood glucose, insulin, proinsulin, C-peptide, and GLPs. Wet weight was defined as the amount of water in the food or stool. Every treatment except placebo assisted in reducing the amount of wet weight, nitrogen, sodium, and potassium in fecal matter but only patients given a treatment containing GLP-2 increased absolute absorption of wet weight and sodium [72]. None of the treatments were found to significantly increase urine volume. Additionally, this study also indicated that specific combinations of GLP-1 and GLP-2 can additively improve small intestinal absorption. All treatments significantly reduced the fecal wet weight, energy, nitrogen, sodium, and potassium losses compared to placebo. Only GLP-2 containing treatments increased absolute absorption of wet weight and sodium and only GLP-1+2 improved hydrational status as evaluated by DEXA increases in the fat mass and calculated total body weight. The study concluded that combinations of GLP-1 and GLP-2 can additively improve small intestinal absorption [72] (Table 1).

### 3.2. Peptide YY

Peptide YY (PYY) is a peptide secreted by ileal L cells in response to nutrient stimulation [73]. It was noticed that SBS patients who retained a colon had normal stool output and liquid secretion whereas patients without a colon had high output and secretion. To test this seven patients without a colon, six with jejunum in continuity with a colon, and 12 normal patients each ate a 640 kcal meal. Blood was drawn 3 h later and then tested for GI hormones. Patients with a colon had a median fasting PYY of 71 pmol/L and a normal postprandial rise in PYY. Patients without a colon had a median fasting PYY of 7 pmol/L and a reduced postprandial peptide YY response. Healthy patients had a median fasting PYY of 11 pmol/L. The levels of motilin, a hormone made by the upper small intestine that speeds gastric emptying, were high in patients without a colon. However, motilin was also seen to be low or normal in patients without a colon and those with the shortest remaining jejunum had the most rapid rates of liquid gastric emptying. This indicates that high motilin was not responsible for the rapid gastric emptying seen in patients without a colon. It was suggested by this study that low PYY may cause rapid gastric emptying of liquid [74]. Treatment for SBS utilizing PYY may be useful for patients and should be investigated further (Table 1).

### 3.3. IGF-1

Following intestinal resection growth hormone (GH) is considered to be one of the major hormones that stimulate intestinal adaptation. One theory notes that this may be due to stimulating the increase of circulating Insulin-like Growth Factor-I (IGF-I) concentrations and levels of IGF-I expressed locally in the bowel. IGF-I will bind to binding proteins (IGFBP) that either prevent or allow it to cross the capillary membrane. When delivered at its target, IGF-I increases protein synthesis in the jejunal muscularis layer along with effects in the jejunal mucosa associated with increases in mucosal DNA, protein, and crypt depth [75]. In a study conducted on rats it was concluded that a co infusion of TPN and IGF-I after massive bowel resection resulted in an increase in the body weight and intestinal mucosal cellularity [76]. This suggests that IGF-I might be useful in inducing intestinal adaptation in human patients. Additionally, ghrelin, a peptide made by the endocrine cells of the intestinal tract, is reported to stimulate the release of GH and IGF-1 [77]. A trial was run to determine the potential therapeutic effectiveness of ghrelin in SBS. Rats with SBS were given TPN along with ghrelin at 10 μg/kg/day. The results were that intestinal adaptation was stimulated by increasing the villus height and crypt depth of the small intestine, especially in the ileum. Surprisingly, the crypt cell proliferation rate increased in the ilium and jejunum but the IGF-1 levels were unchanged. A different study suggested that ghrelin actually works to stimulate intestinal adaptation via a different mechanism. It was suggested that ghrelin promotes intestinal epithelial cell proliferation through a PI3K/Akt pathway and EGFR trans-activation both converging to ERK 1/2 phosphorylation [78]. Regardless, the results of the rat study seemed to indicate that ghrelin may be effective in the treatment of SBS (Table 1).

### 3.4. Thyroid Hormone

Another factor associated with intestinal adaptation following resection is thyroid hormone (TH), consisting of triiodothyronine (T3) and thyroxine (T4). TH is an important trophic hormone that has been found to play a role in proliferation of GI tract cells [79]. In rats with drug induced hypothyroidism, normal levels of the enzymes lactase, sucrase, and intestinal alkaline phosphatase (IAP) were observed to develop slower than in untreated rats [80]. Further, TH treatment in rats which had undergone bowel transection resulted in a 16 g increase in jejunum weight (*p* = 0.001) and 14 g increase in ileum weight (*p* = 0.001) when compared with SBS rats treated with saline. TH treated rats also exhibited 144 μm higher jejunal villus height (*p* = 0.001), 38 μm higher crypt depth (*p* = 0.048) and significantly increased enterocyte proliferation in the jejunum and ileum (*p* = 0.020, *p* = 0.001, respectively) than in those with saline treatment [81].

The exact mechanism through which TH affects the GI tract is not well understood, however TH has previously been found to regulate gene transcription in several other tissues by binding to TH-responsive genes [82,83,84]. In rat models, TH has been found to induce intestinal alkaline phosphatase (IAP) mRNA in differentiated rat villus cells. IAP is an enzyme found on the brush-border of the small intestine specifically expressed by well differentiated villus enterocytes and, as such, effectively acts as a marker for enterocyte differentiation [85]. The precise mechanism through which TH influences the GI system and its potential role as an SBS treatment has yet to be more precisely defined and may be a promising area of study(Table 1).

### 3.5. Cholecystokinin

As referenced earlier, cholestasis, the blockage of bile acid flow, is one of the most common TPN associated hepatic dysfunctions in pediatric SBS. An intriguing area of study is the role of cholecystokinin (CCK) in mitigating the severity of cholestasis and other TPN associated dysfunctions. CCK is a peptide hormone that is secreted by endocrine cells in the proximal small intestine and stimulates gallbladder contraction and promotes small intestinal motility [86]. In several animal studies, administration of cholecystokinin has been found to treat and prevent TPN-associated injury. Daily IV infusions of CCK-octapeptide have been found to prevent TPN-induced gallbladder stasis in prairie dogs, reducing the gallbladder to hepatic bile 3H-cholic acid specific activity ratio (Rsa) by 0.38 (*p* < 0.05), following a 10 day period on TPN, indicating prevention of gallbladder stasis [87]. In rats, twice daily CCK-octapeptide treatment was found to prevent detrimental changes in bile composition and hepatic bile secretion following 7 days of TPN. However, CCK treated rats did not exhibit reductions in hepatic bile flow, taurocholate secretory rate maximums (SRm), or bile salt-independent bile flow when compared to their untreated counterparts [88]. CCK has also been observed to affect pro-inflammatory agent levels in rats who have undergone 80% bowel resection and on TPN. CCK treated rats showed a 6.709 pg/mL decrease (*p* < 0.05) in IL-1 levels and a 4.794 pg/mL decrease (*p* < 0.001) in TNF-alpha levels compared to those only on TPN [89]. Additionally, bile acid secretion was upregulated by CCK in a similar study comparing TPN and TPN plus CCK in piglets [90].

The role of CCK in SBS treatment is not fully defined, however, along with the above results, there have also been studies that show minimal to no improvement with administration of CCK. In 8 week old rabbits over 12 days, CCK treatment failed to significantly improve bile flow or bile acid secretion. There was a slight trend in improvement of bile sulfobromophthalein (BSP) secretion, but these results were not statistically significant [91].

In human studies, CCK has been assessed in its ability to alleviate cholestasis in neonates. High dose CCK treatment in infants who had undergone major surgeries with resulting parenteral nutrition-associated cholestasis resulted in bilirubin level reduction by about 51.3% when compared to those who had received a lower dose but such had no effect on AST or ALT levels [92]. A similar finding resulted following administration of CCK on neonates in which there was 28.5% fewer infants within the treatment group showing direct bilirubin levels of >5.0 mg/dL [93]. CCK was also assessed as a reversing agent of cholestasis. When administered to eight neonates, CCK infusions restored bile flow in 7 of the patients and postoperative isotope biligraphy showed normal excretion to the biliary tree and duodenum [94]. In summary, CCK remains a promising treatment for TPN associated cholestasis, however further research is necessary to establish the peptide has a definitive treatment (Table 1).

### 3.6. Omega-3

Lack of enteral stimulus is a principal mechanism for hepatic and gut injury with SBS patients receiving parenteral nutrition. However, not all components of enteral feedings engage the intestinal mucosa at the same level. Long-chain fatty acids, especially the polyunsaturated fat Omega 3 (ω-3 FA), are implicated in reducing PNALD in children. In male rats, Sukhotnik et al. (2010) demonstrated that an ω-3 FA fish oil–based lipid, administered through an oral or intraperitoneal route resulted in elevated bowel weight, mucosal weight, mucosal cDNA and protein expression, villous height, crypt depth, cell proliferation, and apoptosis relative to the sham animals [95]. However, relative to the SBS group, rats that received oral a ω-3 FA fish oil–based lipid had a lower jejunum bowel and mucosal weight, jejunal villous height, and cell proliferation. Rats that received this lipid intraperitoneally followed the opposite trend, suggesting that the parenteral route provided the strongest intestinal adaptive response [95].

Intravenous administration of ω-3 FA was successfully used to remedy some effects of PNALD in several other studies. In a clinical study of 15 adults that developed cholestasis from soybean oil-based intravenous lipid emulsion (ILE), 12 patients had normalized bilirubin levels after 4 weeks relative to baseline [96]. Based on other clinical results, Diamond et al. predicted that administration of ω-3 FA would improve bile flow, reduce steatosis, and promote an anti-inflammatory state, ultimately preventing the effects of PNALD [97] (Table 1).

### 3.7. TGR-5 Agonists

The Takeda G protein-coupled receptor 5 (TGR5) is expressed in enteroendocrine L cells. A study conducted on STC-1 cells that express TGR5 mRNA in which mRNA interference experiments showed that reduced expression of TGR5 resulted in reduced secretion of GLP [98]. This suggested a relationship between TGR5 and the level of circulating levels of GLP. Two studies were conducted on piglets aimed to determine if TGR5 agonists would be effective at increasing circulating levels of GLP-2. In the first study, a sample size of six parenteral fed piglets per group were gavaged with vehicle, olive extract (OE; 10 or 50 mg/kg), or ursolic acid (UA; 10 mg/kg) and GLP-2 levels were measured for 6 h. UA is a potent TGR5 agonist and the study determined that it was able to induce strong GLP-2 secretion in the piglets. In the second study neonatal pigs were subjected to transection, done as a sham, or 80% mid-small intestine resection to simulate SBS. Two days post procedure these piglets were assigned to treatments for 10 days. Treatments included transection and vehicle, resection and vehicle, resection and 30 mg UA, or resection and 180 mg/kg OE. Plasma GLP-2, intestinal histology, cell proliferation, and gene expression were all measured after 9 days. This second study found that supplementation of UA or OE did not significantly improve growth and proliferation of cells. This was postulated to be due to the stimulating effects of the resection overwhelming any potential enhancing effects of the TGR5 agonists given. They suggest a further study using more potent TGR5 agonists be conducted [99]. Our study from 2015 shows that a TGR5 agonist prevented TPN injury [26] (Table 1).

### 3.8. Rapamycin

Rapamycin, also known as sirolimus, is an immunosuppressive drug frequently used to treat certain types of cancer as well as a therapeutic designed to prevent graft rejection in solid organ transplantation. Specifically, rapamycin functions as an mTOR kinase inhibitor in order to prevent the progression of cell growth in cancer and autoimmune disease, in which mTOR signaling is often upregulated [100]. mTOR is known to regulate biological processes such as translation, ribosome genesis, glucose metabolism, and hypoxic injury response [101]. Rapamycin use is well documented as an immunosuppressive for intestinal transplant in extreme cases of SBS, as well as management of peritoneal sclerosis [15,102,103,104,105,106]. However, in the context of SBS and TPN induced injury, rapamycin has been shown to be useful as a therapeutic for autophagy reduction in PNALD. Autophagy suppression is a well-known mechanism of injury associated with PNALD and parenteral lung injury as mentioned previously in this review. Zhang et al. showed promising results in a rat model where autophagy was induced in rapamycin and TPN rats and then compared to TPN only rats. In the rapamycin group results showed significantly lower levels of reactive oxygen species and reduction in endoplasmic reticulum stress indicators such as binding immunoglobulin protein (BIP) and spliced X-box-binding protein-1 (sXBP1). Additionally, improved hepatic function and less steatosis was noted in the livers of the rapamycin treated rats [107]. Overall, rapamycin has shown significance as a suppressor of autophagy in a wide range of clinical settings related to bowel pathology [107,108] (Table 1).

### 3.9. Chenodeoxycholic Acid (CDCA)

Another novel therapeutic validated for use in SBS or PNALD is chenodeoxycholic acid (CDCA), a dual agonist of the FXR and TGR5 receptors implicated in gut atrophy and bile acid secretion. By restoring bile acids to the gut via duodenal administration of CDCA, significant mitigation of TPN associated injury has been achieved. In a study done on three groups of neonatal piglets administered either enteral feeds, TPN, or TPN and duodenal CDCA, there was significant reduction in markers of cholestasis and steatosis in the CDCA group, as well as induction of GLP-1, GLP-2, and FGF19 when compared to the TPN only group [18]. This same study showed a nearly four-fold improvement in direct bilirubin levels, a normalization of serum triglycerides and bile acids, and a significant increase in weight gain at the end of the study. These results indicate CDCA to be a potent gut trophic agent as well as stimulating factor for GLP secretion. A separate study done by Broughton et al. found that IV administration of CDCA reduced TPN induced pigmented gallstones in prairie dogs. Six prairie dogs given IV CDCA on TPN for roughly 40 days had no gallstones compared to all of the TPN prairie dogs which all had gallstones [109]. Many other studies have found improvements in gut function or hepatoprotection related to TPN injury by addition of other bile acids or components of bile such as taurine, tauroursodeoxycholic acid, or oleanolic acid [26,110,111] (Table 1).

### 3.10. Carbamazepine

In addition to its known role as an anti-epileptic medication, studies investigating non-traditional use of the drug in mitigating complications associated with SBS have found carbamazepine (CBZ) to hold significant therapeutic benefit. CBZ is FDA approved for neurological conditions such as epilepsy and bipolar disorder as it inhibits voltage-gated sodium and calcium channels [112]. In humans, CBZ is metabolized to carbamazepine-10,11-epoxide via the CYP 450 system enzymes CYP3A4 and CYP2B6 [113].

Notably, research on carbamazepine and other anti-epileptic medications in relation to SBS and TPN has focused on their bioavailability, contraindications, and metabolism [114]. CBZ has also been demonstrated to have negative side-effects with long-term usage, such as tissue death in the liver. Evidence of liver damage has been noted from elevated gamma-glutamyltransferase (GGT) and aminotransferase (ALT and AST) levels [115]. These side effects may be due to acute toxicity from accumulating metabolites, hyperactivity of the CYP450 system, or immunological complications [115].

However, CBZ has been implicated in cell-survival mechanisms such as autophagy. Hidvegi et al. (2010) demonstrated that hepatocytes of mice with alpha-1-antitrypsin Z (ATZ) deficiency given supplemental CBZ were able to clear more of the misfolded proteins through autophagy, reducing hepatic load and fibrosis [116]. Thus, CBZ-induced autophagy may pave the path forward for mitigating liver fibrosis seen with long-term TPN usage. Song et al. (2022) introduced supplementary oral carbamazepine (30 mg/kg/day) to piglets given TPN and observed a decrease in hepatic cholestatic deposits with CBZ treatment (12.5 ± 2.4, *p* = 0.038) vs. TPN animals (TPN 15.7 ± 2.2, *p* < 0.001). Additionally, elevated GGT levels in TPN (26.3 ± 6.9 IU/L) decreased when CBZ was introduced (19.8 ± 2.7 IU/L, *p* = 0.047) [117] (Table 1).

## 4. Surgical Management of SBS

Despite the remarkable advance in pharmacotherapeutics for SBS, some patients cannot wean off long-term TPN. Intestinal transplantation can be definitive therapy. Especially for patients with IFALD, liver and small intestine or multivisceral transplantation can be performed. According to the intestinal transplant registry report, 47% of all intestinal transplants were small intestine, 27% were small intestine and liver, and 27% were multivisceral [118]. However, graft survival rate remains particularly low (one year graft survival was 66.1% and five years was 47.8%) [119].

Besides transplantation, some intestinal reconstruction surgeries have been developed. These surgeries focus to slow down intestinal transit, to increase contact time between nutrients and mucosa, to correct remnant bowel dilation and stasis, to improve intestinal motility, and to increase mucosal surface area. Longitudinal intestinal lengthening and tailoring (LILT) and Serial Transverse Enteroplasty (STEP) are the procedures most widely used. LILT was first introduced by Bianchi et al. in 1980. In this technique, a dilated section of small intestine is divided longitudinally into two intestinal halves and then these parts are fashioned together into two narrower tubes, doubling the length of the original intestinal segment [120,121]. Reinshagen et al. reported the results of LITI on 53 patients. Forty-one of the patients survived, and 36 were successfully weaned from PN [122]. LILT is a surgically demanding procedure requiring a healthy mesentery with no fibrosis. STEP was proposed by Kim et al. using a piglet model in 2003. In the STEP procedure, a dilated bowel is alternatively cut with a stapler and then a zig-zag pattern is created [123]. The international STEP Data Registry reported the outcome of STEP on 97 patients; 87 patients survived without intestinal transplantation. Of these 87 survivors, 48 were weaned from PN. Overall outcomes between LILT and STEP are apparently similar [124,125]. The STEP procedure seems to be a relatively easy procedure to perform, as it does not require anastomoses nor mesentery dissection, supporting this technique being accepted more widely [126]. Additionally, there have been previous attempts at novel procedures [127]. Although intestinal transplantation can be the most critical surgical therapy, it would be difficult to perform intestinal transplantation for all SBS patients. Recent developments of surgical technique are remarkable. Further investigations of intestinal reconstruction surgery are needed and expected in the future(Table 1).

## 5. Conclusions

TPN is employed in the absence of enteral nutrition in order to maintain appropriate nutritional support in patients ranging in age from neonates to adults. This lifesaving treatment is associated with a wide range of morbidities and even mortality, therefore it is essential to understand the mechanisms behind TPN associated injury. TPN induces a lack of luminal contents, which is well known to cause gut atrophy and alter liver signaling through the FXR-FGF19 and TGR5-GLP pathways. New research suggests that TPN may even have adverse effects on other organs such as the brain and lung are also under investigation. Currently, hormonal therapeutics, such as GLP, IGF1 and peptide YY are the more well understood treatments for TPN complications. Given the breadth of mechanisms to target and widespread downstream effects, there has also been success seen in therapeutic interventions with other hormones, off-label usages of drugs, and lipids, such as thyroid hormone, cholecystokinin, omega 3, rapamycin, CDCA, and carbamazepine. Additionally, development of novel surgical strategies has shown increasing success at mitigating TPN complications. We now have a growing mechanistic understanding of the systemic effects of TPN and ways to target these mechanisms. As our understanding of PN associated injury continues to develop further investigation into therapeutic targets to mitigate disease and achieve enteral autonomy sooner proves more and more successful.

## Figures and Tables

**Figure 1 nutrients-14-04660-f001:**
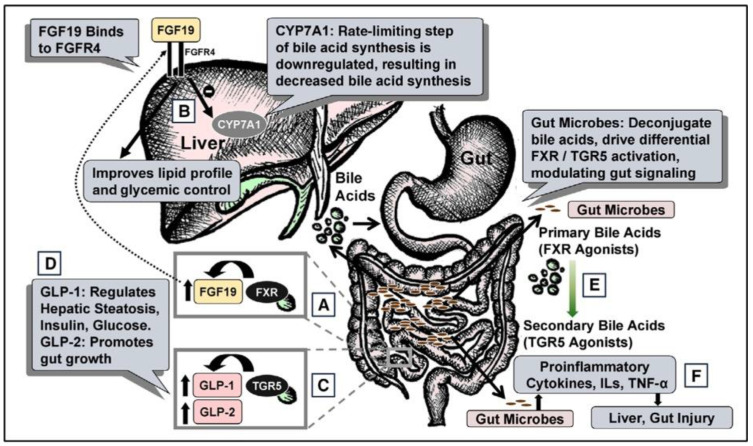
Total Parenteral nutrition Associated injury. (**A**) During regular enteral nutrition, luminal bile acids activate gut farnesoid X receptor (FXR), releasing fibroblast growth factor 19 (FGF19). (**B**) FGF19 regulates hepatic cholesterol 7α-hydroxylase (CYP7A1), fat, and glucose metabolism. (**C**) Gut TGR5 activation by bile acids releases glucagon-like peptide-1 (GLP-1) and GLP-2. (**D**) GLP-1 modulates hepatic steatosis, insulin, and glucose; GLP-2 modulates gut growth. (**E**) Gut microbes deconjugate primary bile acids (FXR agonists) to secondary bile acids (TGR5 agonists). (**F**) Gut microbes also regulate cytokines, interleukins (ILs), and tumor necrosis factor-α (TNF-α), modulating liver and gut injury. FGFR4, FGF receptor 4. “Reproduced with permission from American Society for Parenteral and Enteral Nutrition, Nutrition in clinical practice; published by SAGE Publications, Inc., 2020”.

**Figure 2 nutrients-14-04660-f002:**
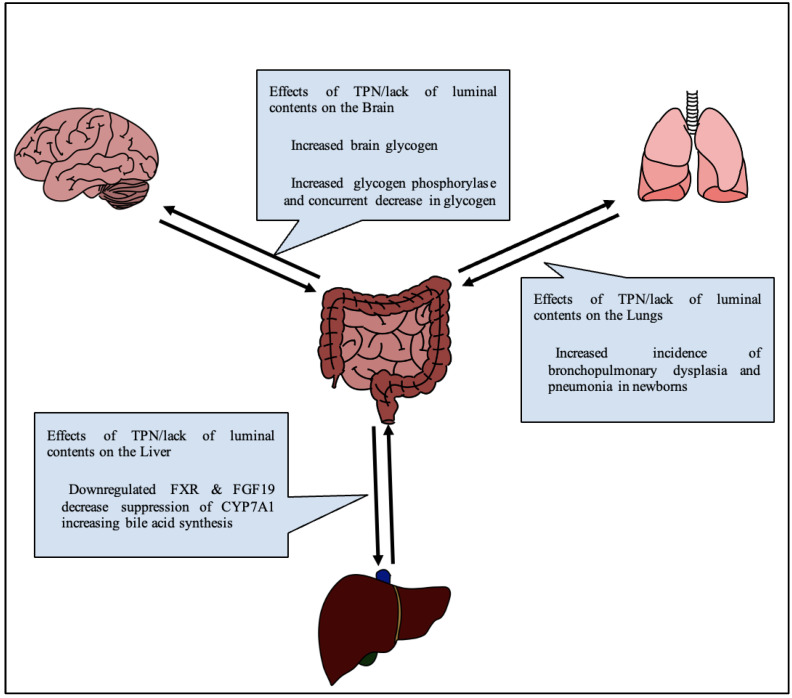
Generalized diagram of effects of TPN on various gut-organ axes.

## Data Availability

Not applicable.

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
