# Peer review of "Novel Therapeutic Approaches for Mitigating Complications in Short Bowel Syndrome"

_nutrients, 2022, doi:10.3390/nu14214660_

Round 1

Reviewer 1 Report

This review features discussions on novel therapeutic approaches for short bowel syndrome (SBS). The small intestine plays an important role in the absorbance of nutrients, and congenital abnormalities or surgical removal of a section of the small intestine damaged the small intestine, resulting in an inability of the small intestine. Total parental nutrition (TPN) compensates small intestine's inability, but TPN has some problems, such as severe liver injury and lack of luminal contents. The authors summarized the effect of TPN on the body and novel therapeutic approaches to mitigation strategies for TPN-induced injury. The following points should be clarified.

The central theme of this manuscript is the novel therapeutic approaches for mitigating complications in SBS. I recommend that the authors summarize what we now know about the therapeutic approaches for mitigating complications in SBS in a table or figure.

Reviewer 2 Report

This review comprehensively introduces the influence of TPN on gut brain, gut lung, and gut liver axis, as well as the treatment of SBS and related complications. It is a very valuable review with rich and organized contents. I have the following questions:

1. What about the key words?

2. What are the effects of short bowel syndrome and TPN on intestinal flora. I think this is also an important part of the gut liver axis.

3. What is the probability of SBS children suffering from mental diseases such as autism? I think SBS and TNP not only affect the neurodevelopment but also affect the occurrence of mental illness.
